# Effectiveness of a service design training program to improve clinical nurses' compassion and problem-solving in Korea

Keum-Seong Jang[1], Mikyoung Lee[2,3], Hyunyoung Park[1], Kyung-Hee Chung[4], Myeong Baek[1], Young-Ran Kweon[1], Yun-Hee Kim[5]*

1 College of Nursing, Chonnam National University, Gwangju, Korea, 2 Department of Nursing, Kwangju Women's University, Gwangju, Korea, 3 Department of Psychology, University of Munich, Munich, Germany, 4 Department of Nursing, Nambu University, Gwangju, Korea, 5 Department of Nursing, Mokpo National University, Muan, Korea

* kimyunhee@mnu.ac.kr

## Abstract

Service design is an innovative tool used to improve the quality of patient experience, therefore, making it necessary for nurses to be able to implement it. The aim of this study was to examine the effects of a training program for patient experience-based nursing service design (PEN-SD) on clinical nurses' compassion and problem-solving ability. This study employed a mixed-methods design: a one-group, pretest-posttest design was used as the quantitative approach, and structured interviews were used as the qualitative approach. The participants were 21 nurses recruited from a university hospital in Korea. A PEN-SD training program was conducted from September 1 to October 6, 2018. After the training program, the participants' compassion significantly improved ($p = .025$) but there was no significant difference in their problem-solving ability ($p = .313$). In the structured interviews, majority of the participants ($n = 17$) felt that they were able to consider problems from the patient's perspective. They also reported being able to identify solutions to problems through careful observation ($n = 5$). The PEN-SD training program was effective in improving compassion among nurses.

## Introduction

Service design is a user-centered service development method that applies design thinking to all processes of service development. It is a useful tool for transforming healthcare services by improving customer experiences through an in-depth and realistic understanding of customers' personal experiences [1, 2]. There is a need to explore both patient engagement from a new perspective in the context of healthcare services and the importance of compassion and empathy for patient experiences among healthcare staff [3]. With this in mind, a service design has recently been applied in the healthcare industry as an innovative tool for designing customer-based service values to improve the quality of patient experience [4]. The method in the service design process is used as an effective problem-solving tool to identify the root causes

**Data Availability Statement:** The data are available from the Dryad database (https://doi.org/10.5061/dryad.7sqv9s4v8).

**Funding:** 1. Jang KS, Kweon YR; No. CRI-17037-21; Chonnam National University Hospital Biomedical Research Institute, Korea; https://bri.cnuh.com/main.php. 2. Park HY; No. 2017R1C1B5018380; National Research Foundation of Korea, Korean government; https://www.nrf.re.kr/index. The funders had no role in study design, data collection and analysis, decision to publish, or preparation of the manuscript.

**Competing interests:** The authors have declared that no competing interests exist.

of, and solutions to, various problems occurring at the service touch-point, from the user's perspective [5].

Service design was first introduced at Maggie's Cancer Caring Centre in the UK in 1996 to create a comfortable hospital environment for patients. Since then, it has been used to improve patient experience in healthcare services [6–8]. A previous study reported the benefits of applying service design to the healthcare field [9]. For example, a radiology department benefited from service design by reducing the waiting period for examinations from 2.6 days to 1.5 days and increasing the number of examinations from 21,078 to 22,236 as well as revisits to the hospital [10]. Existing studies have also reported decreased hospital readmission rates and improved health-related quality of life as a result of applying service design [11–13]. As such, service design has been effective in improving the quality of services in hospitals.

Given that all interested parties in a hospital as well as the patients are involved in the process of service design formulation, it is an innovative method that can ultimately provide integrative services to meet patients' needs. Accordingly, to encourage improvement and development of services in nursing, it is crucial to secure human resources that can implement service design. For patients to experience the best possible nursing service, it is imperative that nurses, who spend most of their time with patients empathize with the patient's experience and develop problem-solving skills. Additionally, service design utilizes different methods in a step-wise approach which necessitates a systematic training program [1, 2]. To the best of our knowledge, no research has been conducted to assess the effectiveness of a nursing service design training program to improve nurses' compassion and problem-solving in Korea.

Considering that service design is a service improvement method that focuses on empathetic skills for enhancing customers' experiences, and problem-solving from the customer's perspective [5], we hypothesized that the patient experience-based nursing service design (PEN-SD) training program would facilitate improvement in nurses' compassion and problem-solving ability. Therefore, we developed this training program and assessed its effectiveness in improving compassion and problem-solving ability.

## Materials and methods

### Research design

This study employed a mixed-methods design, using quantitative and qualitative approaches to investigate the effectiveness of the PEN-SD training program on improving nurses' compassion and problem-solving ability. In particular, the explanatory design was applied; first, we collected and analyzed quantitative data using a one-group, pretest-posttest design to address the research questions, and then conducted structured interviews as the qualitative approach to support the initial quantitative findings [14].

### Participants

The sample size was calculated using the G*power 3.1.9 program. In previous research, where the effectiveness of the program in improving nurses' compassion and problem-solving ability has been assessed [15, 16], the effect size was calculated per variable, which was 0.5–0.6. The effect size for the current research was based on the aforementioned result. The minimum sample size was 20, calculated for analysis using the Wilcoxon signed-rank test based on the aforesaid result with the power of the test $(1-\beta) = 0.8$, effect size $(d) = 0.6$, and significance level $(\alpha) = .05$.

We conveniently sampled 25 participants out of whom only 21 were included in the analysis, after excluding four who responded poorly to the surveys. The final participants—21 nurses working at a university hospital in Korea—were either charge nurses or held positions

high enough to influence changes in nursing services. They understood the purpose of the research and were willing to participate in five weeks of training.

## Instruments

Lee [17] developed a tool comprising 13 questions for measuring compassionate competence: seven questions on communication skills, three questions on sensitivity, and three questions on insightfulness. The questionnaire used a 5-point Likert scale, wherein a higher score signified higher compassion. Cronbach's α was .93 in Lee [17], and .95 in the current study.

Lee et al. [18] also developed a tool for measuring problem-solving ability; it comprises 45 questions: 5 questions on identifying problems, 10 questions on cause analysis, 10 questions on developing alternatives, 10 questions on planning and execution, and 10 questions on evaluating performance. A 5-point Likert scale was used, wherein a higher score signified better problem-solving ability. Cronbach's α in both Lee et al. [18] and this study was .94.

Interview questions, organized by following the advice of seven service design experts, included: "Are there any differences after the training in terms of patient-care-related problems that can occur in the field?," "Are there any methods or tools learned in the course of the program to improve nursing services?," and "What was the most helpful aspect of the PEN-SD training program?"

## Intervention

**Developing the PEN-SD training program.** We developed the PEN-SD training program between March and July 2018, based on the four-stage double diamond process [19] by adding the "understand" and "grow" phases from the six-stage service design [20]. We added these two phases to the widely-used four-stage double-diamond process because the "understand" phase was necessary for orientation and team building to conduct a five-week project. As cooperation through team activities is essential for good project results, it is imperative for team members to acquaint well with each other [20]. Additionally, as a phase for sharing project results and reflecting upon individual and team activities, the "grow" phase is necessary for applying and evaluating solutions derived from service design in the field. The finalized design approach for each step is as follows.

The "understand" phase is the stage to set a goal and plan for the project, thereby comprising a method for preparing the entire process such as team building, creating empathy maps, "How Might We (HMW)" questioning for topic selection, and contextual research. The "discover" phase is about identifying practical problems by forming a consensus with service users, which comprises interviews for on-site surveys and an affinity map, a framework for organizing investigative findings. The "define" phase comprises persona modeling for structuring the results derived from the "discover" phase and defining problems from the customer's perspective, a patient journey map for visualizing the patient experience, and the 5 Whys technique for establishing service goals by finding opportunities and root causes. The "develop" phase comprises brain writing, random word, and idea sketches as part of creating ideas to solve the problems identified. The "deliver" phase is a prototyping stage to review and improve solutions to the problems by creating visual prototypes in different ways to express ideas. The last phase, "grow," involves presenting the results of the service developed or improved and exchanging feedback.

**Implementing the PEN-SD training program.** From September 1 to October 6, 2018, the PEN-SD training program was conducted over five sessions lasting a total of 24 hours. Details are presented in Table 1 and Fig 1. From the "understand" phase to the "deliver" phase, six facilitators coached each team, as needed.

**Table 1. Overview of the patient experience-based nursing service design training program (*n* = 21).**

| SD¹ process | Contents | Duration | Tools |
|---|---|---|---|
| Understand | PEN-SD training program preparation<br>Understanding of oneself and others through empathy, exploration, and selection of topics<br>Contextual research | 3 hours | Brain storming<br>Empathy map<br>Decision grid<br>How Might We? |
| Discover | Field research preparation<br>Field research implementation: observation,<br>FGI, shadowing | 2 hours | Field survey schedule<br>Affinity map |
| | Data organization and analysis using<br>the affinity map | 3 hours | |
| Define | Visualize patient experience | 3 hours | Patient journey map |
| | Create persona | 3 hours | Persona, 5 Whys |
| Develop | Find touch-point and pain-point<br>Establish PEN-SD service-goal<br>Ideation | 3 hours | Brain writing<br>Random word<br>Idea sketch |
| Deliver | Implementing PEN-SD service<br>scenario prototype | 4 hours | Prototyping<br>(Toy blocks, colored<br>-paper) |
| Grow | Presentation and appraisal of PEN-SD<br>training program | 3 hours | |

¹SD: Service design, FGI: Focus group interview

The first session consisted of *Understand* and *Discover* phases (5 hours). In the "Understand" phase (3 hours), one hour of mini lecture was given to introduce the PEN-SD process and to overview this training program. Thereafter, the participants were organized into teams of five–six based on their working hospitals and departments. For topic selection, the participants conducted a field survey through interviewing or shadowing for one hour to identify the actual problem at their current hospital. Then, they used an empathy map to summarize service users' saying, doing, thinking, and feeling and selected a priority topic using a decision grid. Finally, "HMW" questions concerning the actual problems were asked to clarify the topics in a positive and improvable manner. In the "Discover" phase (2 hours), content was created for field research guiding and detailed questions related to "HMW." To solve the problems, the participants devised a field research plan to draw insights by observing the actual environment or activities of the customer who used the service. They also conducted interviews, shadowing, and survey and searched relevant data in journals, articles, or statistics.

The second session included *Discover* and *Define* phases (6 hours). In the "Discover" phase (3 hours), the results from the first session were organized and analyzed using an affinity map.

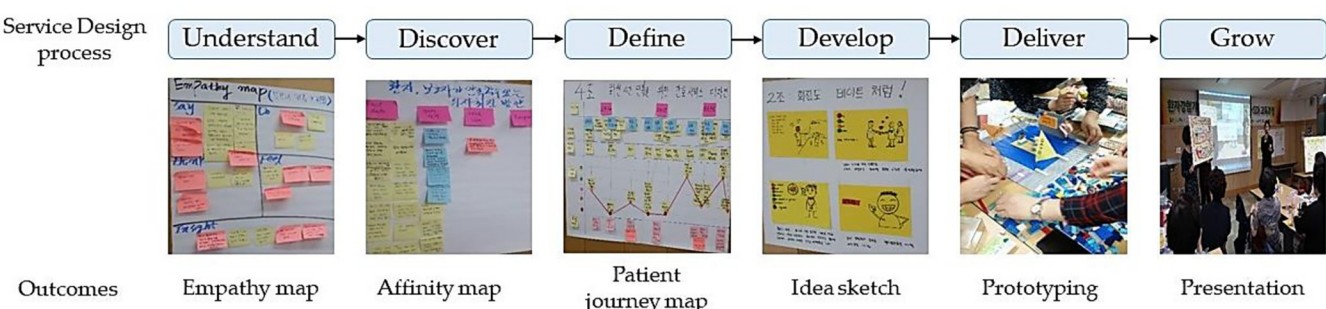

**Fig 1. Example of the PEN-SD training program outcomes (*n* = 21).**

In the "Define" phase (3 hours), a patient experience journey map was created to visualize the patient experience based on the results of the "Discover" phase, and core insights were derived. Finally, a persona representing the service recipients was created to identify the problem in detail.

The third session consisted of *Define* and *Develop* phases (6 hours). In the "Define" phase (3 hours), the 5 Whys technique was employed based on the patient experience journey map and persona created in the second session to identify the fundamental cause. The participants addressed the service needs and discussed the solutions for the patients' pain-points. In the "Develop" phase (3 hours), the discussed ideas were drawn using brain writing or random word. The selected idea was visually outlined via idea sketch to determine how to implement it.

The fourth session was the *Deliver* phase (4 hours). In this phase, the specified ideas from the "Develop" phase were visibly presented through prototyping using a variety of materials, such as toy blocks and colored papers. All teams presented their prototypes, exchanged feedback, and modified these prototypes by reflecting on the feedback.

Finally, the fifth session was the *Grow* phase (3 hours). In this phase, we concluded the program with a presentation session on the complete service results obtained through team activities during the five sessions and the "Deliver" phase.

## Data collection

For the pretest, questionnaires for measuring general characteristics, compassion, and problem-solving ability were administered before commencing the PEN-SD training program on September 1, 2018.

The posttest of the experimental group was conducted using two methods. First, the quantitative approach used compassion and problem-solving ability questionnaires on October 6, 2018, after completing the PEN-SD training program. Second, the qualitative approach used the structured interview method to support the quantitative findings. Three months after the PEN-SD training program, from January 7 to 25, 2019, individual interviews were conducted regarding the PEN-SD training program experience. The survey was conducted only with 21 participants who agreed to the interview, as confirmed by researchers who had engaged with these participants during the training. Those who could not personally attend the interview responded to a structured interview questionnaire that was received by them in an envelope.

## Data analysis

The collected data were analyzed using the IBM SPSS/WIN 25.0 program. Participants' general characteristics were analyzed as percentage, mean, and standard deviation. Differences in the compassion and problem-solving ability of the experimental group before and after training were analyzed using the Wilcoxon signed-rank test. Content analysis was performed to analyze the interview data following the content analysis procedure by Graneheim and Lundman [21]. First, the interview transcripts were repeatedly read to comprehend the meaning as a whole. Then, they were classified into meaning units based on the content and context. This was conducted independently by two researchers, and the classification was discussed until both researchers reached an agreement. Subsequently, common types were identified, and similar items were categorized. In this process, the results of content analysis were derived through repeated review and discussion.

## Ethical consideration

This study was conducted after obtaining the approval of the Biomedical Research Ethics Committee at Chonnam National University Hospital (CNUH-2018-182). We also obtained

the developer's approval via email for using the study tools. Those who agreed to participate in the survey completed a questionnaire after receiving a written or verbal explanation of the purpose of the research, anonymity, and withdrawal from participation. Informed consent was obtained from the participants beforehand for audio-recording of the interview. If they did not agree to audio-recording, the interview was recorded on paper.

## Results

### Participants' general characteristics

The average age of the participants was 48.00 (± 3.61) years. -Among them, 13 (39.4%) held a master's degree. The average duration of clinical experience was 26.61 (± 3.43) years. In terms of position, there were 13 head nurses (61.9%) and eight charge nurses (38.1%). Nine nurses (42.9%) were working in the surgical unit (Table 2).

**Quantitative analysis of changes in clinical nurses' compassion and problem-solving.** Comparing the effects of the PEN-SD training program before and after the training indicated that compassion significantly improved from the median score of 4.12 at pretest to 4.35 at posttest ($Z = -2.25$, $p = .025$). The results of each subscale were as follows: communication ($Z = -2.42$, $p = .016$), sensitivity ($Z = -1.41$, $p = .159$) and insight ($Z = -0.51$, $p = .608$), indicating a significant improvement in communication but not in sensitivity and insight (Table 3).

There were no significant differences in problem-solving before and after the training (median scores 3.73 and 3.82 [$Z = -1.01$, $p = .313$]), respectively. The results of each subscale were as follows: There was no significant difference in problem clarification ($Z = -1.44$, $p = .150$), cause analysis ($Z = -0.26$, $p = .793$), development of alternative ($Z = -0.28$, $p = .778$), planning/execution skill ($Z = -1.41$, $p = .158$), and performance assessment ($Z = -1.26$, $p = .209$) before and after the training (Table 3).

**Qualitative analysis of changes in clinical nurses' compassion and problem-solving.** According to the content analysis using participants' responses to the interview questions as keywords, their responses to the first question, "Are there any differences after the training in

**Table 2. Demographic characteristics of the participants ($n$ = 21).**

| Variables | Categories | M ± SD or n (%) |
|---|---|---|
| Age (years) | | 48.00 ± 3.61 |
| Education | Diploma | 4 (23.8) |
| | RN-BSN | 2 (9.5) |
| | Master | 13 (39.4) |
| | Doctor | 2 (9.5) |
| Clinical experience (years) | | 26.61 ± 3.4 |
| Position | Head nurse | 13 (61.9) |
| | Charge nurse | 8 (38.1) |
| Work Unit | Medical unit | 6 (28.6) |
| | Surgical unit | 9 (42.9) |
| | OBGY[1] unit | 1 (4.8) |
| | ICU[2] | 2 (9.5) |
| | OR[3]/RR[4] | 3 (14.3) |

[1]OBGY: Obstetric gynecology

[2]ICU: Intensive care unit

[3]OR: Operation room

[4]RR: Recovery room

**Table 3. Effect of patient experience-based nursing service design training program on compassion and problem-solving ability (*n* = 21).**

| Variables | Categories | pretest | posttest | Z (*p*) |
|---|---|---|---|---|
| | | Median (IQR) | | |
| Compassion | Communication | 4.13 (3.37–4.38) | 4.38 (3.71–4.63) | -2.42 (.016) |
| | Sensitivity | 4.20 (3.90–4.60) | 4.60 (3.90–4.80) | -1.41 (.159) |
| | Insight | 4.00 (3.63–4.00) | 4.00 (3.50–4.13) | -0.51 (.608) |
| | Total | 4.12 (3.64–4.29) | 4.35 (3.72–4.50) | -2.25 (.025) |
| Problem-solving ability | Problem clarification | 3.80 (3.50–4.10) | 4.00 (3.70–4.20) | -1.44 (.150) |
| | Causal analysis | 3.70 (3.40–3.95) | 3.70 (3.50–4.00) | -0.26 (.793) |
| | Development of alternative | 3.80 (3.35–3.90) | 3.70 (3.45–4.00) | -0.28 (.778) |
| | Planning/execution skills | 3.60 (3.25–3.95) | 3.90 (3.45–4.05) | -1.41 (.158) |
| | Performance assessment | 3.90 (3.55–4.10) | 3.90 (3.60–4.30) | -1.26 (.209) |
| | Total | 3.73 (3.49–3.97) | 3.82 (3.57–4.03) | -1.01 (.313) |

terms of patient-care-related problems?" were divided into two categories: more empathy with patients and coworkers and changes in problem-solving process, with nine main responses (Table 4).

Analysis of answers to "More empathy with patients and coworkers" revealed that the most common response was, "Thought from a patient's point of view in the field" (n = 17). A few examples of participants' responses are as follows:

**Table 4. Participants' responses after patient experience-based nursing service design training program (*n* = 21).**

| Questions | Categories | Responses | n* |
|---|---|---|---|
| Are there any differences after the training in terms of patient-care-related problems? | More empathy with patients and coworkers | Thought from a patient's perspective in the nursing field | 17 |
| | | Became more observant of patients | 6 |
| | | Helped communicating with ward nurses | 1 |
| | Changes in problem-solving process | Put more efforts to find a way to solve the problem through careful observation | 5 |
| | | Used 5 Whys to take a more fundamental and objective approach to solve a problem | 3 |
| | | Used journey mapping to find the cause of the problem in the ward | 2 |
| | | Looked for new ways rather than solve problems in an existing way | 3 |
| | | Collected and categorized problems | 2 |
| | | Tried to approach the root cause of the problems objectively and find fundamental solutions | 2 |
| | No response | | 6 |
| Are there any methods you applied in the clinical nursing field after the program? | 5 Whys technique | To explore underlying causes when consulting with new nurse members | 1 |
| | User shadowing | To observe patients unfolded a way to approach nursing | 1 |
| | Never applied | I was too busy to apply new methods It was difficult to apply the methods alone | 19 |
| What was the most helpful aspect of the program? | Using the 5 Whys technique | It was an opportunity to solve problems fundamentally It allowed looking at problems from different perspectives | 17 |
| | Patient journey mapping | It was a different approach from before to find the cause of the problem, when there was a problem with patient care in the ward | 1 |
| | Creating persona models | It helped figuring out undiscovered problems via interview | 1 |
| | No response | | 2 |

* Duplicated responses.

*"It made me think a lot, 'Why is that person reacting like this in this situation?' and I tried to understand the patient or caregiver." (*Participant 21)

*"I was able to consider patients and caregivers first in terms of improving the ward system, and I am trying to change the way of thinking, 'What if I were the patient or caregiver. . .'." (Participant 3)*

For "Changes in problem-solving process," "Tried to find ways to solve the problem through careful on-site observation" was the most common response (n = 5), and three participants responded that they "Put an effort to find the cause of the problem objectively and fundamentally using the 5 Whys." A few examples of participants' responses are as follows:

*"After the training, now I could do a deep dive to solve patients' problems. By doing this, I was able to produce key insights by synthesizing various situations from different perspectives." (Participant 17)*

*"Since the PEN-SD training program, I have been trying to find the root causes of problems in the nursing field using the '5 Whys'." (Participant 9)*

Regarding the second question, "Are there any methods you applied in the clinical nursing field after the program?", three categories were extracted: 5 Whys technique, user shadowing, and never applied. The never applied category had the most responses (n = 19) with two common reasons, "I was too busy to apply this to the field" and "It was difficult to apply the methods alone". A few examples of participants' responses are as follows:

*"Too many things to manage at work, just no time. . . I couldn't apply it to the real situations." (Participant 1)*

*"It is very difficult for the nursing department alone to try new methods; we need substantial support from other departments such as the administration and management departments. Without their support, it would be hard to apply in reality." (Participant 16)*

As for the third question, "What was the most helpful aspect of the program?", three categories were revealed: using the 5 Whys technique, patient journey mapping, and creating persona models. Most participants responded that the 5 Whys technique helped them think about fundamental problem-solving (n = 17). An example of participants' responses is as follows:

*"When solving problems in clinical situations, we usually focus on solving visible issues; however, this training allowed us to think about fundamental areas in problem-solving by focusing on the '5 Whys'." (Participant 18)*

## Discussion

In this study, compassion among nurses significantly improved after the PEN-SD training program. This result is consistent with Eines et al.'s [22] study, where service design was applied in nursing homes for 17 nurses, resulting in an improvement of their compassion. Our participants conducted a field survey using interviews and shadowing techniques during the "Understand" and "Discover" phases. A patient's needs must be humanely examined to identify the subjective meaning of their perspective and experience; this will improve the compassion of nurses [23, 24]. We believe that contextual, in-depth interviews and close observations are effective in promoting nurses' compassion toward their patients.

Koo [25] identified that interview skills, scenarios, and duration of interviews are important to fully empathize with others and to enhance compassion. Our participants conducted hour-long interviews in hospitals and in the field. However, there was insufficient pre-interview training on interview skills and the kind of questions to prepare. This should be considered in future PEN-SD training programs. For interviews to play a meaningful role in improving compassion, it is necessary to organize a training session on how to conduct an interview, expected scenarios, and techniques for obtaining authentic answers [26].

Additionally, Koo [25] suggested that the duration of interviews must be at least two hours to sufficiently elicit the participant's experiences. Therefore, it is necessary to allocate at least two hours for an interview when planning a nursing service design program in future.

There was no significant improvement in the participants' problem-solving ability after the PEN-SD training program. Although their problem-solving ability did not improve in the quantitative data, we identified—through their responses—perceptual changes related to problem-solving in the qualitative data: "Put more efforts to find a way to solve the problem through careful observation," "Use the 5 Whys to take a more fundamental and objective approach to solve a problem," and "Look for new ways rather than solve problems in an existing way." In particular, participants mentioned that the 5 Whys technique was the most useful method they learned during the PEN-SD training program. The 5 Whys is a technique to identify the root cause of a problem by asking a "why" question five times, and is used in root-cause analysis for problem-solving [27]. These results indicate that the PEN-SD program improved the participants' problem-solving ability to some degree. For more significant improvements in future, it is necessary to simplify the methods and toolkits used in each stage of the PEN-SD training program and modify it to acquire a systematic and comprehensive process for problem-solving.

Given that service design is a method that applies design thinking to all aspects of a service to improve the overall customer experience through in-depth understanding of customer experience, problem-solving, therefore, ultimately means improving the overall user experience [1, 2]. During the PEN-SD training program, as a method to identify—and resolve—the root-cause of problems by identifying customer needs, participants were educated regarding the use of different methods in addition to field interview, empathy map, affinity map, patient journey map, persona, and the 5 Whys technique. However, it might have been difficult for the participants unfamiliar with service design to grasp the essence of the service design approach as a problem-solving process, as they had to focus on completing the tasks required for each step within a limited time frame. Therefore, in order to improve participants' problem-solving abilities, future programs should consider repeated training sessions so that participants can skillfully handle and internalize the service design method. Moreover, a strategy to ensure adequate field-exposure to directly apply the service design method across diverse service fields is also required. We believe that extending the nursing service design training period—for example, to eight weeks—will reduce the pressure on nurse participants. Thus, it is necessary to allocate extra time and support for step-by-step approaches; practical cooperative experience through exchange of feedback, ideas, and alternative proposals; and prototyping to develop the nursing service design training program.

Finally, it should be noted that this research was an innovative attempt to introduce service design in the nursing field. This study facilitates the provision of quality services of nurses who empathize with the patients' experiences by reflecting upon their needs, at a time when such experiences are included as a measure of the performance of medical services. However, one of the limitations of this study includes the single-group pretest-posttest experimental design. In future studies, it will be necessary to reconfirm the effects of the PEN-SD training on

compassion and problem-solving ability among nurses through a control-group pretest-post-test design.

## Conclusions

We implemented a PEN-SD training program for nurses as a method for developing a patient experience-based nursing service and examined the effectiveness of the program on nurses' compassion and problem-solving ability. The PEN-SD training program lasted for a total of 24 hours through five sessions. We found that there was a significant improvement in compassion among the nurse participants. We also found that the PEN-SD training experience changed the nurses' perspective as they reported assessing problems from the patient's perspective with empathy and also reported trying to identify the root-cause of problems through close observation. We hope that the PEN-SD training program developed in this study can be used as an approach to improve the service design competencies required for developing the best nursing services reflecting the experiences of patients. Based on the present results, we suggest the following. First, our study utilized a one-group pre-posttest design; thus, repetitive investigations should be conducted in the future to verify the effect of the PEN-SD program on nurses' compassion and problem-solving using a control-group pre-posttest design. Second, it is necessary to develop a PEN-SD-method tool tailored for nurses for widespread application of the PEN-SD in various nursing fields. Third, we expect to see a variety of service improvement and development studies in the nursing field reflecting the needs of patients by utilizing the PEN-SD method developed in this study.

## Author Contributions

**Conceptualization:** Keum-Seong Jang, Mikyoung Lee, Hyunyoung Park, Kyung-Hee Chung, Myeong Baek, Young-Ran Kweon, Yun-Hee Kim.

**Data curation:** Mikyoung Lee, Yun-Hee Kim.

**Formal analysis:** Mikyoung Lee, Yun-Hee Kim.

**Funding acquisition:** Keum-Seong Jang, Hyunyoung Park, Young-Ran Kweon.

**Investigation:** Mikyoung Lee, Kyung-Hee Chung, Yun-Hee Kim.

**Methodology:** Keum-Seong Jang, Hyunyoung Park, Kyung-Hee Chung, Myeong Baek.

**Project administration:** Mikyoung Lee.

**Resources:** Keum-Seong Jang, Young-Ran Kweon.

**Software:** Myeong Baek, Yun-Hee Kim.

**Supervision:** Keum-Seong Jang.

**Validation:** Keum-Seong Jang, Mikyoung Lee, Hyunyoung Park, Kyung-Hee Chung, Myeong Baek, Young-Ran Kweon, Yun-Hee Kim.

**Writing – original draft:** Yun-Hee Kim.

**Writing – review & editing:** Mikyoung Lee, Hyunyoung Park, Yun-Hee Kim.

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
