## [Decision Letter · Decision Letter 0]

11 Jan 2022

PONE-D-21-21797Implementing a Service Design Training Program to Improve Clinical Nurses’ Compassion and Problem-solving: A Mixed-Methods Pilot StudyPLOS ONE

Dear Dr. Yun-Hee Kim,

Thank you for submitting your manuscript to PLOS ONE. After careful consideration, we feel that it has merit but does not fully meet PLOS ONE’s publication criteria as it currently stands. Therefore, we invite you to submit a revised version of the manuscript that addresses the points raised during the review process.

Please submit your revised manuscript by February 25, 2022. If you will need more time than this to complete your revisions, please reply to this message or contact the journal office at plosone@plos.org. Please include the following items when submitting your revised manuscript:A rebuttal letter that responds to each point raised by the academic editor and reviewer(s). You should upload this letter as a separate file labeled 'Response to Reviewers'.A marked-up copy of your manuscript that highlights changes made to the original version. You should upload this as a separate file labeled 'Revised Manuscript with Track Changes'.An unmarked version of your revised paper without tracked changes. You should upload this as a separate file labeled 'Manuscript'.

We look forward to receiving your revised manuscript.

Kind regards,

Sharon Mary Brownie

Academic Editor

PLOS ONE

Journal Requirements:

Additional Editor Comments 

Reviewers have provided recommendations to improve your manuscript. Please consider each in detail and respond accordingly.

Reviewers' comments:

Reviewer's Responses to Questions

**Comments to the Author**

1. Is the manuscript technically sound, and do the data support the conclusions?

Reviewer #1: Partly

Reviewer #2: Yes

Reviewer #3: Yes

Reviewer #4: Partly

2. Has the statistical analysis been performed appropriately and rigorously? 

Reviewer #1: Yes

Reviewer #2: Yes

Reviewer #3: Yes

Reviewer #4: N/A

3. Have the authors made all data underlying the findings in their manuscript fully available?

Reviewer #1: Yes

Reviewer #2: No

Reviewer #3: Yes

Reviewer #4: No

4. Is the manuscript presented in an intelligible fashion and written in standard English?

Reviewer #1: Yes

Reviewer #2: Yes

Reviewer #3: Yes

Reviewer #4: Yes

5. Review Comments to the Author

Reviewer #1: Title: Implementing a Service Design Training Program to Improve Clinical Nurses’ Compassion and Problem-solving: A Mixed-Methods Pilot Study

Comment 1: It lacks specify of study area, period which makes it un attractive to some extent. Ho you see this issue?

Abstract: Dear Authors thank you for you work. Under this section the aim that you proposed is completely different from the title. As your title is about” Implementing a Service Design Training Program to Improve Clinical Nurses’ Compassion and Problem-solving: A Mixed-Methods Pilot Study” however you’re in your abstract it talks about specific program” line 28-30. The aim of this study was to examine the effects of a training program for patient experience-based nursing service design (PEN-SD) on clinical nurses’ compassion and problem”

Comment 2: How you can Negotiate these two different concepts here?

Comment 3: you have written the sentences on Line 37-38. “This research is the first attempt to introduce a service design in the nursing field.” Under result section of your abstract. How many data base you have searched to conclude the service design in nursing is first?

If you are sure and able to respond the above concern, please take the sentence to introduction section your abstract. If not, this sentences your personal saying which will not be scientifically sounding

Comment 4: Your abstract section lacks the method through which you passed to come up with the result that exist here. What will be your possible justification for this?

Generally I am not satisfied with the existing abstract unless it is improved.

Comment 5: Introduction line 51-53: is talking about Exploring patient engagement, however the title is about service design. As the introduction about first impression for reader it has to start with sentence talking about service design

Comment 6: line 59, 73 and 83 are you talking about Method or what you stated “Methodologies” which is not appropriate for your content.

Comment 7: line 72-88 were not cited elsewhere. How you will justify it?

Comment 8: result section was well organized however table and information with table were incomplete. i.e table are vague including title needs further improvement.

Comment 9: line 319 onward needs scientific evidence for possible justification.

Comment 10: 403-417/Conclusion of the document needs further improvement as it is not inline with the objective and finding of the study.

Comment 11. All references of the document needs further consideration for the format and to make the reference full.

Reviewer #2: First of all, I want to appreciate the authors for such novel idea. I do have few concerns and I forward as follow.

1. You used "pretest-posttest design" for the quantitative part. However, the study design used for the qualitative part is appropriately mentioned. "structured interviews" written in your document is not a design. hence, you have to incorporate a qualitative study design best fits for the study; I think phenomenological study design best fits for your qualitative part.

2.Which mixed design is applied in your paper? Triangulation Design, the Embedded Design, the Explanatory Design, or the Exploratory Design. Better to mention the exact mixed design applied.

3. Though explanation of the intervention is good, to much explanation makes the manuscript large. So better to reduce the information from line 126 up to 214.

4. I think it is better to separate the qualitative part and develop a theory using a Grounded theory design for further researches.

Reviewer #3: The manuscript is technically sound. it touches on quality of health care and patient satisfaction with services which is an area of interest globally

I have only a few minor observations to make:

1) I find the data analysis section to be shallow and too brief as it mainly talks about what analysis done but not much on how that analysis was done. It will be good to see a detailed description of how the analysis was actually done

2) It is not clear from the text the number of participants who took part in the interviews. Line 225-227 says "The survey was conducted only with those participants who agreed to the interview, as confirmed by researchers who had engaged with these participants during the training. " Is it possible to indicate exactly how many participated?

3) Line 248: I think the words "of them" present a grammatical error and needs another look

Reviewer #4: It seems to be reference 16; the article of this dissertation was published in 2016 and includes 17 items.

Lee Y, Seomun G. Development and validation of an instrument to measure nurses' compassion competence. Applied Nursing Research. 2016 May 1; 30: 76-82.

Which qualitative approach did you use?

Explain the results of the qualitative approach further

6. PLOS authors have the option to publish the peer review history of their article (what does this mean?). If published, this will include your full peer review and any attached files.

Reviewer #1: **Yes: **Getahun Fetensa

Reviewer #2: No

Reviewer #3: No

Reviewer #4: No

---

## [Author Response · Author response to Decision Letter 0]

27 Apr 2022

*We also attached the file "Responses to Reviewer Comments"

Responses to Reviewer 1

Thank you very much for your letter of revision suggestions and the opportunity to revise our manuscript entitled “Implementing and Evaluating a Service Design Training Program to Improve Clinical Nurses’ Compassion and Problem-solving in Korea: A Mixed-Methods Pilot Study” (PONE-D-21-21797). Your comments proved to be helpful for revising the manuscript. Below we described how we responded to your comments in blue and highlighted the changes using the track changes mode in the manuscript. Thank you.

Title: Implementing a Service Design Training Program to Improve Clinical Nurses’ Compassion and Problem-solving: A Mixed-Methods Pilot Study

Comment 1: It lacks specify of study area, period which makes it unattractive to some extent. How you see this issue?

We appreciate your constructive feedback. We have now added the study place in Korea in the present title. Since the title is already long, the study period was described in the Abstract and the Materials and Methods instead of the title part. Thank you for your understanding.

Abstract

Dear Authors thank you for your work. Under this section the aim that you proposed is completely different from the title. As your title is about” Implementing a Service Design Training Program to Improve Clinical Nurses’ Compassion and Problem-solving: A Mixed-Methods Pilot Study” however you’re in your abstract it talks about specific program” line 28-30. The aim of this study was to examine the effects of a training program for patient experience-based nursing service design (PEN-SD) on clinical nurses’ compassion and problem”

Comment 2: How you can Negotiate these two different concepts here?

Thank you for the excellent point. We agree with you that there was a discrepancy between the title and the abstract content. We have now revised both Title and Abstract to keep consistency between them as follows (p. 1, lines 1-2 & p. 2, lines 28-32):

“Title: Implementing and Evaluating a Service Design Training Program to Improve Clinical Nurses’ Compassion and Problem-solving in Korea: A Mixed-Methods Pilot Study”

“The aim of this study was to implement a training program for patient experience-based nursing service design (PEN-SD) and examine the effects of the program on clinical nurses’ compassion and problem-solving ability.”

Comment 3: you have written the sentences on Line 37-38. “This research is the first attempt to introduce a service design in the nursing field.” Under result section of your abstract. How many data base you have searched to conclude the service design in nursing is first? If you are sure and able to respond the above concern, please take the sentence to introduction section your abstract. If not, this sentences your personal saying which will not be scientifically sounding.

Considering your comment, we changed this sentence using the word innovative instead of first as follows (p. 2, lines 40-41): 

“This research is an innovative attempt to introduce a service design in the nursing field.”

Comment 4: Your abstract section lacks the method through which you passed to come up with the result that exist here. What will be your possible justification for this? Generally I am not satisfied with the existing abstract unless it is improved.

We have now added the following sentence to explain the method in the Abstract (p. 2, lines 31-32):

“This study employed mixed methods: a one group, pretest-posttest design was used as the quantitative approach and structured interviews as the qualitative approach.” 

Introduction

Comment 5: Introduction line 51-53: is talking about Exploring patient engagement, however the title is about service design. As the introduction about first impression for reader, it has to start with sentence talking about service design.

Thank you for the thoughtful suggestion. We now moved the sentence introducing the service design into the first sentence and reorganized the whole paragraph as follows (p. 3, lines 51-61):

“Service design is a user-centered service development method that applies design thinking to all processes of service development; it is a useful tool for transforming healthcare services by improving customer experiences through an in-depth and realistic understanding of customers’ personal experiences [1,2]. There is a need to explore patient engagement from a fresh perspective in the context of healthcare services, as well as a need to explore the importance of compassion and empathy for patient experiences among healthcare staff [3]. With this in mind, a service design has been recently applied in the healthcare industry as an innovative tool for designing customer-based service values to improve the quality of patient experience [4]. The method in the service design process is used as an effective problem-solving tool to identify—from users’ perspectives—root causes of, and solutions to, various problems occurring at the service touch-point [5].”

Comment 6: line 59, 73 and 83 are you talking about Method or what you stated “Methodologies” which is not appropriate for your content.

Reflecting your comment, we now used the word either method or approach instead of methodology in the whole manuscript. Thank you.

Comment 7: line 72-88 were not cited elsewhere. How you will justify it?

We are grateful to your thorough review. We now properly cited the information we used as follows (p. 4, lines 79-80 & lines 83-86):

“Additionally, service design utilizes different instrumental methods step by step, which necessitates a systematic training program [1,2].”

“Considering that service design is a service improvement method that focuses on empathetic skills for enhancing customer experience, and problem-solving from customer’s perspective [5], the patient experience-based nursing service design (PEN-SD) training program will facilitate improvement in nurses’ compassion and problem-solving ability.”

Results 

Comment 8: result section was well organized however table and information with table were incomplete. i.e table are vague including title needs further improvement.

To improve clarity of the Tables 3 and 4, we changed the Table titles as follows and revised information in Table 4 (please refer to p. 13 & pp. 16-17):

“Table 3. Effect of Patient Experience-based Nursing Service Design Training Program on Compassion and Problem-solving Ability (N=21)”

“Table 4. Participants’ Responses after Patient Experience-based Nursing Service Design Training Program”

Discussion

Comment 9: line 319 onward needs scientific evidence for possible justification.

To support our discussion based on scientific evidence, we have now added citations of previous studies as follows (p. 17, lines 334-339 & p. 20, lines 390-393): 

“In this study, a six-stage process for the PEN-SD training program was developed by adding the “understand” and “grow” phases to the existing four-stage process [20]. In the “understand” phase—the first step of the program—participants selected the project topic, confirmed the matters to be solved during this training, and discussed the approach direction. This is because the PEN-SD training program was organized to decide the project topic to work on for five weeks and solve the problems through the service design process [20].”

“Given that service design is a method that applies design thinking to all aspects of a service to improve the overall customer experience through in-depth understanding of customer experience, problem-solving, therefore, ultimately means improving the overall user experience [1,2 ].”

Conclusions

Comment 10: 403-417/Conclusion of the document needs further improvement as it is not in line with the objective and finding of the study.

We really appreciate your insightful feedback to improve the quality of our manuscript. We have now revised the Conclusion section to properly reflect the objectives and findings of the study as well as suggestions as follows (p. 21, lines 419-436):

“We implemented a PEN-SD training program for nurses as a method for developing a patient experience-based nursing service and examined the effectiveness of the program on nurses’ compassion and problem-solving ability. The PEN-SD training program lasted for a total of 24 hours through five sessions. We found that there was a significant improvement in compassion among the participating nurses. We also found that the PEN-SD training experience changed the nurses’ perspective as they reported assessing problems from the patient’s perspective with empathy and also reported trying to identify the root-cause of problems through close observation. We hope that the PEN-SD training program developed in this study can be used as an approach to improve the service design competencies required for developing the best nursing services reflecting the experiences of patients. Based on the present results, we suggest the following. First, our study utilized a one group pre-posttest design; thus, repetitive investigations should be conducted in the future to verify the effect of the PEN-SD program on nurses’ compassion and problem-solving using a control group pre-posttest design. Second, it is necessary to develop a PEN-SD-method tool tailored for nurses for widespread application of the PEN-SD in various nursing fields. Third, we expect to see a variety of service improvement and development studies in the nursing field reflecting the needs of patients by utilizing the PEN-SD method developed in this study.”

References

Comment 11. All references of the document need further consideration for the format and to make the reference full.

We have now checked the references thoroughly and revised the format according to the guidelines. Thank you.

Responses to Reviewer 2

Thank you very much for your letter of revision suggestions and the opportunity to revise our manuscript entitled “Implementing and Evaluating a Service Design Training Program to Improve Clinical Nurses’ Compassion and Problem-solving in Korea: A Mixed-Methods Pilot Study” (PONE-D-21-21797). Your comments proved to be helpful for revising the manuscript. Below we described how we responded to your comments in blue and highlighted the changes using the track changes mode in the manuscript. Thank you.

First of all, I want to appreciate the authors for such novel idea. I do have few concerns and I forward as follow.

1. You used "pretest-posttest design" for the quantitative part. However, the study design used for the qualitative part is appropriately mentioned. "structured interviews" written in your document is not a design. hence, you have to incorporate a qualitative study design best fits for the study; I think phenomenological study design best fits for your qualitative part.

We really appreciate your constructive feedback. We realized that we did not provide sufficient information on our study design. Following your suggestions, we described the study design in detail in our reply to your next comment 2 together. Thank you for your understanding. 

2. Which mixed design is applied in your paper? Triangulation Design, the Embedded Design, the Explanatory Design, or the Exploratory Design. Better to mention the exact mixed design applied.

Thank for your valid comments and questions (Comments 1 & 2). We applied the explanatory design as the mixed design. We described the details in the Research Design section as follows (p.4, lines 92-97): 

“This study employed mixed methods using quantitative and qualitative approaches to investigate the effectiveness of the PEN-SD training program on nurses’ compassion and problem-solving ability. In particular, the explanatory design was applied; first, we collected and analyzed quantitative data using a one group, pretest-posttest design to address research questions, and then conducted structured interviews as the qualitative approach to support the initial quantitative findings [14].”

3. Though explanation of the intervention is good, too much explanation makes the manuscript large. So better to reduce the information from line 126 up to 214.

Following your suggestion, we reduced some information throughout the Intervention section (pp. 6-10). Thank you.

4. I think it is better to separate the qualitative part and develop a theory using a Grounded theory design for further researches.

Thank you for the valuable suggestion. We totally agree with you. Future research will benefit from separating the qualitative part and developing a theory using a grounded theory design. 

Responses to Reviewer 3

Thank you very much for your letter of revision suggestions and the opportunity to revise our manuscript entitled “Implementing and Evaluating a Service Design Training Program to Improve Clinical Nurses’ Compassion and Problem-solving in Korea: A Mixed-Methods Pilot Study” (PONE-D-21-21797). Your comments proved to be helpful for revising the manuscript. Below we described how we responded to your comments in blue and highlighted the changes using the track changes mode in the manuscript. Thank you.

The manuscript is technically sound. It touches on quality of health care and patient satisfaction with services which is an area of interest globally. I have only a few minor observations to make:

1) I find the data analysis section to be shallow and too brief as it mainly talks about what analysis done but not much on how that analysis was done. It will be good to see a detailed description of how the analysis was actually done.

We appreciate your insightful feedback. We agree that the important details on how the analysis was done were missing in the Data Analysis section. We have now added more details on the analysis procedure as follows (p.11, lines 238-245):

“Content analysis was performed to analyze interview data following the content analysis procedure by Graneheim and Lundman [21]. First, the interview transcripts were repeatedly read to comprehend the meaning as a whole. Then, they were classified into meaning units based on the content and context. This was conducted independently by two researchers, and the classification was discussed until both researchers reached an agreement. After that, common types were discovered and similar items were categorized. In this process, the results of content analysis were derived through repeated review and discussion.” 

2) It is not clear from the text the number of participants who took part in the interviews. Line 225-227 says "The survey was conducted only with those participants who agreed to the interview, as confirmed by researchers who had engaged with these participants during the training. " Is it possible to indicate exactly how many participated?

Thanks to you, we realized that we missed the information on the exact number of the participants. We have now added 21 participants in the original sentence as follows (p. 11, lines 229-231):

“The survey was conducted only with 21 participants who agreed to the interview, as confirmed by researchers who had engaged with these participants during the training.”

3) Line 248: I think the words "of them" present a grammatical error and needs another look.

We changed Of them to Among them recommended by a professional editor (refer to p. 12, line 258). We also had the revised manuscript proofread by a professional editor to improve the clarity of the sentences. Thank you.

Responses to Reviewer 4

Thank you very much for your letter of revision suggestions and the opportunity to revise our manuscript entitled “Implementing and Evaluating a Service Design Training Program to Improve Clinical Nurses’ Compassion and Problem-solving in Korea: A Mixed-Methods Pilot Study” (PONE-D-21-21797). Your comments proved to be helpful for revising the manuscript. Below we described how we responded to your comments in blue and highlighted the changes using the track changes mode in the manuscript. Thank you.

1. It seems to be reference 16; the article of this dissertation was published in 2016 and includes 17 items. Lee Y, Seomun G. Development and validation of an instrument to measure nurses' compassion competence. Applied Nursing Research. 2016 May 1; 30: 76-82.

We are thankful to your thorough review. We are aware that Lee’s (2014) dissertation was published in 2016 with 17 items. We used the original instrument with 13 items in Lee’s (2014) dissertation, however. Thank you for your understanding. 

2. Which qualitative approach did you use? Explain the results of the qualitative approach further.

Thank you for your insightful question. We found that sufficient explanation on the qualitative approach was missing in the original manuscript. We have now added more details in the Research Design and Data Analysis sections as follows (p.4, lines 92-97 & p.11, lines 238-245):

“This study employed mixed methods using quantitative and qualitative approaches to investigate the effectiveness of the PEN-SD training program on nurses’ compassion and problem-solving ability. In particular, the explanatory design was applied; first, we collected and analyzed quantitative data using a one group, pretest-posttest design to address research questions, and then conducted structured interviews as the qualitative approach to support the initial quantitative findings [14].”

“Content analysis was performed to analyze interview data following the content analysis procedure by Graneheim and Lundman [21]. First, the interview transcripts were repeatedly read to comprehend the meaning as a whole. Then, they were classified into meaning units based on the content and context. This was conducted independently by two researchers, and the classification was discussed until both researchers reached an agreement. After that, common types were discovered and similar items were categorized. In this process, the results of content analysis were derived through repeated review and discussion.” 

In addition, we described more details on the results of the qualitative approach in the Results section as follows (p. 14, lines 288-292 & p. 15, lines 310-314 & p. 16, lines 320-323):

“According to content analysis using participants’ responses to the interview questions as keywords, their responses to the first question, “Are there any differences after the program in the point of view for the patient-care-related problems?” were divided into two categories: more empathy with patients and coworkers and changes in problem-solving process with nine main responses (Table 4).”

“Regarding the second question, “Are there any methods you applied in the clinical nursing field after the program?”, three categories were extracted: 5 Whys technique, user shadowing, and never applied. The never applied category had the most responses (n = 19) with two common reasons, “I was too busy to apply this to the field” and “It was difficult to apply the methods alone”.

“As for the third question, “What was the most helpful in the program?”, three categories were revealed: using the 5 Whys technique, patient journey mapping, and creating persona models. Most participants responded that the 5 Whys technique helped them think about fundamental problem-solving (n = 17).”

---

## [Decision Letter · Decision Letter 1]

26 May 2022

PONE-D-21-21797R1Implementing and Evaluating a Service Design Training Program to Improve Clinical Nurses’ Compassion and Problem-solving in Korea: A Mixed-Methods Pilot StudyPLOS ONE

Dear Dr.Yun-Hee Kim,

Thank you for submitting your manuscript to PLOS ONE. After careful consideration, we feel that it has merit but does not fully meet PLOS ONE’s publication criteria as it currently stands. Therefore, we invite you to submit a revised version of the manuscript that addresses the points raised during the review process.

We look forward to receiving your revised manuscript.

Kind regards,

Sharon Mary Brownie

Academic Editor

PLOS ONE

Journal Requirements:

Reviewers' comments:

Reviewer's Responses to Questions

**Comments to the Author**

1. If the authors have adequately addressed your comments raised in a previous round of review and you feel that this manuscript is now acceptable for publication, you may indicate that here to bypass the “Comments to the Author” section, enter your conflict of interest statement in the “Confidential to Editor” section, and submit your "Accept" recommendation.

Reviewer #1: All comments have been addressed

Reviewer #2: (No Response)

Reviewer #3: All comments have been addressed

2. Is the manuscript technically sound, and do the data support the conclusions?

Reviewer #1: Yes

Reviewer #2: Yes

Reviewer #3: Yes

3. Has the statistical analysis been performed appropriately and rigorously? 

Reviewer #1: No

Reviewer #2: Yes

Reviewer #3: Yes

4. Have the authors made all data underlying the findings in their manuscript fully available?

Reviewer #1: Yes

Reviewer #2: Yes

Reviewer #3: Yes

5. Is the manuscript presented in an intelligible fashion and written in standard English?

Reviewer #1: Yes

Reviewer #2: Yes

Reviewer #3: Yes

6. Review Comments to the Author

Reviewer #1: The manuscript needs a major English language edition. In another way, the manuscript title and aim were not consistent. I kindly recommend the authors make it consistent. There is a need for sentence re-arrangement in the introduction and abstract sections of the work.

Reviewer #2: Almost all of the concerns raised in the previous revision are well addressed. However, further minor revisions are needed.

#1. As the study is an interventional study (before-after or quasi experimental) the word implementing on the title has no significance importance and the title can be written as “Effectiveness of a service design training program to improve clinical nurses’ compassion and problem-solving skill in Korea” or “Evaluating a service design training program to improve clinical nurses’ compassion and problem-solving skill in Korea”. In both titles implementation of the intervention is already there.

#2. Under the participants heading. In line 100 you mentioned 21 participants were included in the study. in the same paragraph in line 108 the calculated sample size was 20. Why such discrepancies happened? In addition, though there is no means to calculate the sample size for qualitative study, “information saturation” as a basis to determine the sample size. However, nothing was said about it in the method section.

#3. Remove the bullets in lines 169, 182, 188, 195 and 200; simple separate the steps of the interventions in paragraphs

#4. Table and figure headings must be self-explanatory: in line 248 include the area, year and total sample size in bracket. line 167 too.

#5. In the discussion; information written from line 320-343 are more of explanation of the method section. Hence, better to minimize this information.

Reviewer #3: All my concerns have been addressed by the authors and therefore have no further questions to raise.

I recommend the revised manuscript for publication.

7. PLOS authors have the option to publish the peer review history of their article (what does this mean?). If published, this will include your full peer review and any attached files.

Reviewer #1: **Yes: **Getahun Fetensa

Reviewer #2: No

Reviewer #3: No

---

## [Author Response · Author response to Decision Letter 1]

2 Jun 2022

Thank you very much for your constructive feedback and the opportunity to revise our manuscript entitled “Effectiveness of a Service Design Training Program to Improve Clinical Nurses’ Compassion and Problem-solving in Korea” (PONE-D-21-21797R1). Your comments proved to be helpful for revising the manuscript. Below we described how we responded to your comments and highlighted the changes using the track changes mode in the manuscript. Thank you.

Responses to Reviewer 1

The manuscript needs a major English language edition. In another way, the manuscript title and aim were not consistent. I kindly recommend the authors make it consistent. There is a need for sentence re-arrangement in the introduction and abstract sections of the work.

Comment 1: there are many improvements throughout the manuscript. However, English language and grammar needs major revision still. 

Response: Thank you. We had the revised manuscript proofread by a professional editor to improve the clarity of the manuscript. Revisions are track-changed in the manuscript. 

Comment 2: Dear authors thank you for all your effort in revising your manuscript. However, I have a concern regarding the title and aim of the study. Do you think that they are the same?

“Evaluating a Service Design Training Program to Improve Clinical Nurses’ Compassion and Problem-solving in Korea” vs “examine the effects of the program on clinical nurses’ compassion and problem-solving ability”

Response: Following your recommendation, to make the title and aim consistent we revised the title as “Effectiveness of a Service Design Training Program to Improve Clinical Nurses’ Compassion and Problem-solving in Korea”. We think this title includes both implementing and evaluating of the program, and it is consistent with the aim: “The aim of this study was to implement a training program for patient experience-based nursing service design (PEN-SD) and examine the effects of the program on clinical nurses’ compassion and problem-solving ability.” (in the Abstract). Thank you for your understanding. 

Responses to Reviewer 2

Almost all of the concerns raised in the previous revision are well addressed. However, further minor revisions are needed.

#1. As the study is an interventional study (before-after or quasi experimental) the word implementing on the title has no significance importance and the title can be written as “Effectiveness of a service design training program to improve clinical nurses’ compassion and problem-solving skill in Korea” or “Evaluating a service design training program to improve clinical nurses’ compassion and problem-solving skill in Korea”. In both titles implementation of the intervention is already there.

Response: As you recommended, we have now revised the title as “Effectiveness of a Service Design Training Program to Improve Clinical Nurses’ Compassion and Problem-solving in Korea”. This title indeed includes both implementing and evaluating of the program. Thank you.

#2. Under the participants heading. In line 100 you mentioned 21 participants were included in the study. in the same paragraph in line 108 the calculated sample size was 20. Why such discrepancies happened? In addition, though there is no means to calculate the sample size for qualitative study, “information saturation” as a basis to determine the sample size. However, nothing was said about it in the method section.

Response: We really appreciate your insightful feedback. The calculated sample size of 20 was the minimum number of the participants we needed. We also acknowledge that information saturation is a basis to determine the sample size in a qualitative study. Since we conducted structured interviews as the qualitative approach to support the initial quantitative findings, we collected data from the 21 nurses who participated in the program. Thank you for your understanding.

#3. Remove the bullets in lines 169, 182, 188, 195 and 200; simple separate the steps of the interventions in paragraphs.

Response: We have now removed the bullets and instead described each step of the interventions in paragraphs.

#4. Table and figure headings must be self-explanatory: in line 248 include the area, year and total sample size in bracket. line 167 too.

Response: We have now added the sample size information (n = 21) in the headings of all Tables and Figures. Thank you. 

#5. In the discussion; information written from line 320-343 are more of explanation of the method section. Hence, better to minimize this information.

Response: Thanks to you, we realized that some information in the discussion was explaining the method part. Following your suggestion, we revised the discussion section as follows (pp. 17-18, Lines 328-354): 

“In this study, a six-stage process for the PEN-SD training program was developed by adding the “Understand” and “Grow” phases to the existing four-stage process [20]. In the “Understand” phase—the first step of the program—participants selected the project topic, confirmed the matters to be solved during this training, and discussed the approach direction. It was a necessary phase for participants to establish the goals and plans of the project, before they actually discovered the problems. In future programs, it would be useful to organize the “Understand” phase based on the level of the participants so as to increase the effectiveness of the programs. 

The “Grow” phase—the last stage of the PEN-SD training program—was originally designed to help participants apply and evaluate solutions derived through service design in the field, assess how to further develop them based on what they had learned, and suggest future activities [20]. However, in our pilot study applying the PEN-SD training program for a total of 24 hours, it was difficult to directly apply the project outcomes to the nursing clinical field. Future PEN-SD training programs require modifications so that participants can reflect and grow by directly applying the project outcomes to the field. Furthermore, it is imperative to keep the outcomes feasible in future programs with support from hospitals.”

Responses to Reviewer 3

All my concerns have been addressed by the authors and therefore have no further questions to raise. I recommend the revised manuscript for publication.

Response: Thank you very much.

---

## [Decision Letter · Decision Letter 2]

14 Jun 2022

PONE-D-21-21797R2Effectiveness of a Service Design Training Program to Improve Clinical Nurses’ Compassion and Problem-solving in KoreaPLOS ONE

Dear Dr. Yun-Hee Kim,

Thank you for submitting your manuscript to PLOS ONE. After careful consideration, we feel that it has merit but does not fully meet PLOS ONE’s publication criteria as it currently stands. Therefore, we invite you to submit a revised version of the manuscript that addresses the points raised during the review process.

We look forward to receiving your revised manuscript.

Kind regards,

Sharon Mary Brownie

Academic Editor

PLOS ONE

Journal Requirements:

Reviewers' comments:

Reviewer's Responses to Questions

**Comments to the Author**

Reviewer #1: All comments have been addressed

Reviewer #2: All comments have been addressed

Reviewer #3: (No Response)

2. Is the manuscript technically sound, and do the data support the conclusions?

Reviewer #1: Yes

Reviewer #2: Yes

Reviewer #3: Yes

3. Has the statistical analysis been performed appropriately and rigorously? 

Reviewer #1: Yes

Reviewer #2: Yes

Reviewer #3: Yes

4. Have the authors made all data underlying the findings in their manuscript fully available?

Reviewer #1: Yes

Reviewer #2: Yes

Reviewer #3: Yes

5. Is the manuscript presented in an intelligible fashion and written in standard English?

Reviewer #1: Yes

Reviewer #2: Yes

Reviewer #3: No

6. Review Comments to the Author

Reviewer #1: Accept. The authors have addressed the raised comments from my side in two rounds of the review process.

Reviewer #2: (No Response)

Reviewer #3: The manuscript has improved a lot but it is still not in standard English.

Secondly, I feel the discussion is heavy on the development of the program while giving less attention to assessment of effectiveness of the program. I suggest that the authors try to strike a balance or alternatively drop the development bit and only deal with assessing the effectiveness in this manuscript. If they take this route they can develop another manuscript to take care of the development bit. That way this manuscript will be clearer. To support this point, you can see that in the results section there is hardly anything about development of the program, which makes it strange then when the discussion is heavy with development. Further, by describing development of the program in the methods section, it already looks like the intention of the authors in this manuscript was to report on the effectiveness of the program only.

7. PLOS authors have the option to publish the peer review history of their article (what does this mean?). If published, this will include your full peer review and any attached files.

Reviewer #1: **Yes: **Getahun Fetensa (Department of Nursing, Institute of Health Sciences, Wollega University and Department of Health Behavior and society, faculty of Public health, Jimma University)

Reviewer #2: No

Reviewer #3: No

---

## [Author Response · Author response to Decision Letter 2]

7 Jul 2022

Pleased see the uploaded file "Responses to reviewer comments". Thank you.

---

## [Decision Letter · Decision Letter 3]

1 Aug 2022

Effectiveness of a Service Design Training Program to Improve Clinical Nurses’ Compassion and Problem-solving in Korea

PONE-D-21-21797R3

Dear Dr.Yun-Hee Kim

We’re pleased to inform you that your manuscript has been judged scientifically suitable for publication and will be formally accepted for publication once it meets all outstanding technical requirements.

Kind regards,

Sharon Mary Brownie

Academic Editor

PLOS ONE

Reviewers' comments:

Reviewer's Responses to Questions

**Comments to the Author**

1. If the authors have adequately addressed your comments raised in a previous round of review and you feel that this manuscript is now acceptable for publication, you may indicate that here to bypass the “Comments to the Author” section, enter your conflict of interest statement in the “Confidential to Editor” section, and submit your "Accept" recommendation.

Reviewer #1: All comments have been addressed

Reviewer #2: All comments have been addressed

Reviewer #3: All comments have been addressed

2. Is the manuscript technically sound, and do the data support the conclusions?

Reviewer #1: Yes

Reviewer #2: Yes

Reviewer #3: Yes

3. Has the statistical analysis been performed appropriately and rigorously? 

Reviewer #1: Yes

Reviewer #2: Yes

Reviewer #3: Yes

4. Have the authors made all data underlying the findings in their manuscript fully available?

Reviewer #1: Yes

Reviewer #2: (No Response)

Reviewer #3: Yes

5. Is the manuscript presented in an intelligible fashion and written in standard English?

Reviewer #1: Yes

Reviewer #2: Yes

Reviewer #3: Yes

6. Review Comments to the Author

Reviewer #1: All comments were addressed the publication of the manuscript will benefit the journal. the work was concluded in a good manner.

Reviewer #2: (No Response)

Reviewer #3: Most of the issues I raised have been addressed and so have no further comments as far as this manuscript is concerned.

7. PLOS authors have the option to publish the peer review history of their article (what does this mean?). If published, this will include your full peer review and any attached files.

Reviewer #1: **Yes: **Getahun Fetensa

Reviewer #2: No

Reviewer #3: No

---

## [Editor Report · Acceptance letter]

4 Aug 2022

PONE-D-21-21797R3 

Effectiveness of a Service Design Training Program to Improve Clinical Nurses’ Compassion and Problem-solving in Korea 

Dear Dr. Kim:

I'm pleased to inform you that your manuscript has been deemed suitable for publication in PLOS ONE. Congratulations! Your manuscript is now with our production department. 

Kind regards, 

on behalf of

Professor Sharon Mary Brownie 

Academic Editor

PLOS ONE